# Generating Multimodal Metaphorical Features
# for Meme Understanding

## ABSTRACT

Understanding a meme is a challenging task, due to the metaphorical information contained in the meme that requires intricate interpretation to grasp its intended meaning fully. In previous works, attempts have been made to facilitate computational understanding of memes through introducing human-annotated metaphors as extra input features into machine learning models. However, these approaches mainly focus on formulating linguistic representation of a metaphor (extracted from the texts appearing in memes), while ignoring the connection between the metaphor and corresponding visual features (e.g., objects in meme images). In this paper, we argue that a more comprehensive understanding of memes can only be achieved through a joint modelling of both visual and linguistic features of memes. To this end, we propose an approach to generate Multimodal Metaphorical feature for Meme Classification, named MMMC. MMMC derives visual characteristics from linguistic attributes of metaphorical concepts, which more effectively convey the underlying metaphorical concept, leveraging a text-conditioned generative adversarial network. The linguistic and visual features are then integrated into a set of multimodal metaphorical features for classification purpose. We perform extensive experiments on a benchmark metaphorical meme dataset, MET-Meme. Experimental results show that MMMC significantly outperforms existing baselines on the task of emotion classification and intention detection. Our code and dataset are available at https://github.com/liaolianfoka/MMMC.

## CCS CONCEPTS

• **Computing methodologies** → **Natural language processing**; **Computer vision representations**.

## KEYWORDS

meme understanding, metaphor, multimodal

**ACM Reference Format:**
Anonymous submission. 2024. Generating Multimodal Metaphorical Features for Meme Understanding. In *Proceedings of the 32nd ACM International Conference on Multimedia (MM '24), October 28-November 1, 2024, Melbourne, VIC, Australia* ACM, New York, NY, USA, 9 pages. https://doi.org/10.1145/nnnnnnn.nnnnnnn

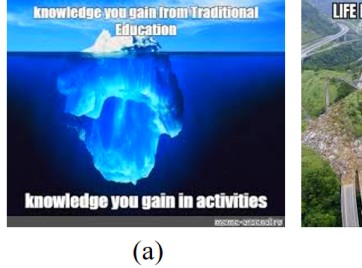

**Figure 1: Accurate interpretation of metaphorical information is crucial to understanding memes**

## 1 INTRODUCTION

Memes, commonly in the form of an image and accompanied text, are known as cultural elements that are widely spread among social media platforms (e.g., Twitter and Reddit) [5]. It has been shown that strong correlation exists between memes and the emotions of those users who produce and share these memes [8]. Therefore, improving computational understanding of memes is considered as of great importance to social media data analytic, with the potential of yielding further benefits to various tasks that are heavily reliant on social media data, such as personalized recommendation, question answering, and open-domain dialog systems.

Recently, various meme classification tasks have been proposed, including emotion classification [23], intention detection [30], and offensive detection [12]. However, performing meme classification is challenging due to its intricacy: memes often convey metaphorical concepts (e.g., humorous, satirical, and symbolic meanings) that require careful interpretation to understand the actual emotions behind. Moreover, metaphorical information is conveyed as multimodal data involving both textual and visual information. Figure 1(a) illustrates an example, where negative emotion is expressed by metaphorizing *knowledge* as *iceberg* and comparing the size of iceberg between *traditional education* and *activities*. In addition, *knowledge* is presented in text while *iceberg* is presented in image. If a model ignores the implicit metaphorical message "*iceberg is knowledge*", it will be difficult to understand that the image is conveying *negative* emotion. Similarly, in Figure 1(b), the negative view that "*2020 is a crack of life*" is expressed by metaphorizing *2020* as a *crack* in *highway*. *2020* is presented in text while *crack* is presented in image – if a model ignores the metaphorical message "*2020 is crack*", this meme is more likely to be classified as *neutral*.

To address the above issue, recent studies have attempted to add human-annotated metaphor to improve models' understanding of meme. Xu et al. [30] produced a meme dataset containing rich metaphors, called MET-Meme, and showed that meme understanding can be enhanced by adding human-annotated metaphors as extra features of memes. In MET-Meme, a metaphor consists of a source concept (e.g., knowledge in Fig. 1) and a target concept (e.g.,

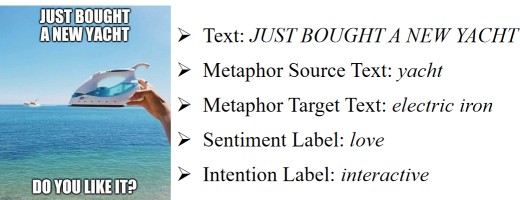

➢ Text: *JUST BOUGHT A NEW YACHT*
➢ Metaphor Source Text: *yacht*
➢ Metaphor Target Text: *electric iron*
➢ Sentiment Label: *love*
➢ Intention Label: *interactive*

**Figure 2: A sample metaphorical meme in MET-Meme, which contains the metaphorical message "*electric iron is yacht*".**

iceberg). MET-meme includes manually annotated source/target concepts. Figure 2 presents an example. The meme contains the metaphorical message "*electric iron is yacht*", where the metaphor source concept is *yacht* and the metaphor target concept is *electric iron*. Besides, Hwang and Shwartz [11] also created a metaphorical meme dataset called MemeCap, and conducted experiments on meme caption generation. However, these studies mainly focus on formulating the linguistic representations of metaphors. In this paper, we show that visual features can provide complementary information to linguistic features of a metaphorical message, and that a multimodal representation of metaphor can improve more accurate meme understanding [26, 32]. This is because understanding a metaphorical message often requires linking multiple concepts together: a model can only effectively link the "entity" in a meme text and "object" in a meme image, if the model understands the common appearance of a textual entity. Similarly, in Figure 1, a model will not even link the iceberg in meme image with the text "iceberg", let alone noticing its size difference above and in water, unless the general appearance of iceberg is provided to the model.

Hence, we propose an approach for multimodal metaphorical meme classification (MMMC), which generates multimodal metaphorical features to improve meme understanding. MMMC generates visual features from textual metaphorical features to obtain multimodal features of memes. In particular, we develop a generative adversarial network (GAN), which maps the textual description of a metaphor concept in a meme to a visual feature space. Thereafter, MMMC feeds the meme with its multimodal metaphorical features for classification, where a multi-stage feature integration method is deployed. We conduct experiments on the benchmark MET-Meme dataset. Experimental results show that MMMC significantly outperforms existing baseline methods on the emotion classification and intention detection tasks. The contribution of this paper can be summarized as:

- To the best of our knowledge, MMMC is the *first* model that generates multimodal metaphorical features to improve meme understanding.
- We design a novel meme classification model that integrates a meme and its multimodal metaphorical features for enhanced classification accuracy.
- We evaluate our method on the MET-Meme dataset. The experimental results demonstrate the effectiveness of our method MMMC. Our source code and data are released for knowledge sharing.

## 2 RELATED WORK

### 2.1 Metaphor

Metaphor is a figure of speech that compares a person or object to something else with similar characteristics. It adds vividness and imagination to descriptions by going beyond literal interpretation(e.g., "*time is money.*"). In this literary device, the concept being described is the target concept (e.g., time), while the concept used to describe it is the source concept (e.g., money). Visual metaphors are also becoming more popular with the advancement of multimedia technology, especially in mediums like memes and posters. Visual metaphors capitalize on the visual medium's ability to convey complex ideas concisely and impactfully.

Previous studies on metaphor mainly focused on linguistic metaphor and made significant progress in identifying, interpreting, and generating metaphors. Choi et al. [4] use contextualized word representations and linguistic metaphor identification theories to detect linguistic metaphors. Chen et al. [3] use out-of-domain data and idioms as additional knowledge to enhance metaphor interpretation. Stowe et al. [27] generate metaphors by encoding conceptual mappings between cognitive domains.

Recently, there has been a growing interest in the study of visual metaphors. Akula et al. [1] create a metaphorical poster dataset and introduce four tasks related to visual metaphors, which include visual metaphor classification, localization, understanding, and generation. Similarly, Zhang et al. [33] develop a metaphorical poster dataset called MultiMET and conduct experiments to empirically demonstrate the beneficial impact of visual metaphors on comprehending posters. Additionally, Stowe et al. [27] utilize large language models and diffusion models to generate visual metaphors.

### 2.2 Meme Classification Methods

Most studies on meme classification are non-metaphorical. They usually adopt a multimodal learning approach, where many effective methods for feature extraction, feature fusion, and multi-task learning have been proposed. For feature extraction, past work relies on LSTM to extract linguistic representation from OCR text [7, 36] while others suggest the usage of different image encoders to extract visual representation [2, 28]. For feature fusion, Guo et al. [9] fuse the textual and visual representations through a weighted combination of all modalities. Koutlis et al. [14] design a dual stage modality fusion module that incorporates the external knowledge. Nguyen et al. [20] design a fusion network which includes a combination of a multi-hop attention network and a stacked attention network. For multi-task learning, Lee and Shen [15] leverage the correlation between tasks and implement four different multi-task network heads having different level of interactions. Moreover, Duan and Zhu [7] adopt modified offline-gradient-blending strategy[29] to alleviate overfitting; Zhu et al. [35] explore meme classification in zero-shot setting; Hazman et al. [10] extract the spatial position of visual objects, faces, and text clusters to assist in meme classification. However, non-metaphorical methods ignore the metaphorical information in the meme, which plays an important role in meme understanding.

On the other hand, some studies put forward metaphorical methods. Xu et al. [30] consider the metaphor source concept and metaphor

target concept as distinct entities and encode them separately. They also demonstrate the potential of incorporating additional metaphorical information to enhance meme comprehension. Additionally, Zhang et al. [34] propose a metaphorical alignment task to improve meme understanding. They employ a conditional generative approach to capture metaphorical analogies and utilize a disentangled contrastive matching mechanism to preserve contextual sensitivity. However, these studies primarily focus on the linguistic representation of metaphor and ignore the visual representation, which is also crucial for metaphor comprehension. Hence, we adopt a different approach that generates visual metaphorical features to enrich the metaphorical representation.

## 3 PROBLEM DEFINITION

In this paper, we propose an approach to automatically generate multimodal features from memes for enhanced understanding of memes' semantic meanings. We showcase the effectiveness of the proposed feature generation method on two meme classification tasks, namely emotion classification and intention detection. Throughout the paper, we use the term *metaphor source text*/*metaphor target text* to represent the textual description of the metaphor source/target concept within a meme metaphor.

Let $m$ be a meme. We represent its features as a four-tuple:

$$m = (o_c, o_v, s_c, t_c) \tag{1}$$

where $o_v$ is the original meme image, $o_c$ is the original text obtained through optical character recognition (OCR), $s_c$ is its metaphor source text, and $t_c$ is its metaphor target text. In the rest of the paper, we use the subscript $v$ and $c$ to distinguish the visual and textual features, respectively. Note that $s_c$ and $t_c$ can be *null* if a meme does not contain metaphorical information.

For a given meme $m$, the emotion/intention classification task is to map $m$ to its ground-truth label from a set of predefined classes [1].

## 4 METHODOLOGY

In this section, we detail our proposed framework, MMMC, which autonomously generates metaphorical features from memes for enhanced meme understanding. Figure 3(a) presents an overview of MMMC. It consists of two phases: *Visual metaphorical feature Generation (VG)* which is to generate visual metaphorical features about metaphors from their corresponding textual descriptions; and *Meme Classification (MC)* which is the integration of multimodal meme features for meme classification.

In VG phase, the textual metaphorical features $s_c$ and $t_c$ extracted from one meme $m$ are mapped to visual metaphorical features via a GAN-based model. Consequently, the original meme $m$ is transformed into $\hat{m}$ with enriched visual features:

$$\hat{m} = (o_v, o_c, s_c, t_c, s_v, t_v) \tag{2}$$

where $s_v$ and $t_v$ are the metaphor features newly introduced and represent the metaphor source image and metaphor target image,

respectively. The set of metaphorical features: $(s_v, s_c, t_v, t_c)$ constitute multimodal metaphorical features to be used in the later phase.

In MC phase, the multimodal metaphorical features of $\hat{m}$ are combined and integrated to jointly perform classification tasks. Next, we describe these two phases in detail.

## 4.1 Visual Metaphorical Feature Generation (VG)

Let $M = \{m_1, m_2, ..., m_n\}$ be a metaphorical meme set with $n$ memes. We denote the entity involved in a meme metaphor text as the metaphor concept (e.g., "yacht" or "electric iron" in Fig 3). For each metaphor concept in $M$, the VG phase aims to generate a metaphor image as its visual metaphorical feature, through a GAN-based mapping model that transforms the textual data to visual feature space.

To achieve this, we first extract all source and target metaphor concepts from $M$ and construct a metaphor concept set containing only unique concepts: $U = \{c_1, c_2, ..., c_{n_c}\}$, where $n_c$ is the number of unique metaphor concepts. Due to the large size of $U$, we divide $U$ into $n_c/l$ subsets, where the $k$-th subset ($k \in \{1, 2, ..., n_c/l\}$) can be denoted as:

$$U_k = \{c_{l(k-1)+1}, c_{l(k-1)+2}, ..., c_{lk}\} \tag{3}$$

For each subset, we train a GAN model that is capable of generating visual features from textual description of a given metaphor concept. We illustrate the overall workflow of the model in Figure. 3(b), and describe the procedure of obtaining such a model as follow.

*4.1.1 Real Images Collection.* First, real-world images of all textual metaphor concepts are collected to serve as the training data for model training. In particular, we collect 50 real images from Internet for each metaphor concept. Our collection criteria are that these images should be diverse, publicly available, and semantically consistent with the textual metaphor concept. For example, given the meme concept *"bread"*, we collect 50 images covering different types of bread using image search engines, and no images of the rock band "Bread". Subsequently, we compile a collection of image-text pairs where in each metaphorical concept is paired with its corresponding visual representation:

$$S_k = \{(r, c) | c \in U_k\} \tag{4}$$

where $r$ is a real image of $c$, and $|S_k| = 50 \times |U_k|$.

*4.1.2 Metaphor Image Generation.* We propose to use a GAN-based model for visual feature generation. Specifically, we train a text-conditional generator $G_t$ and a discriminator $D$, where $G_t$ maps a given piece of text to its visual representation, while $D$ is a classification model which determines if an image is realistic representation for the text. Given a metaphor text $c$, a textual encoder is firstly used to encode $c$ into a feature vector that captures its semantic meaning. The feature vector is subsequently fed into the generator $G_t$ to obtain its visual representation $f$:

$$f = G_t(\Phi(c)) \tag{5}$$

where $\Phi$ is a text encoder. The generated image $f$ by $G_t$ forms a image-text pair with the text input $c$. Together with previously collected set $S_k$, the generated pseudo pair is fed into the discriminator

---

[1]The labels of emotion classification include: *happiness, love, anger, sorrow, fear, hate,* and *surprise.* The labels in intention detection include: *interactive, expressive, purely entertaining, offensive,* and *other*

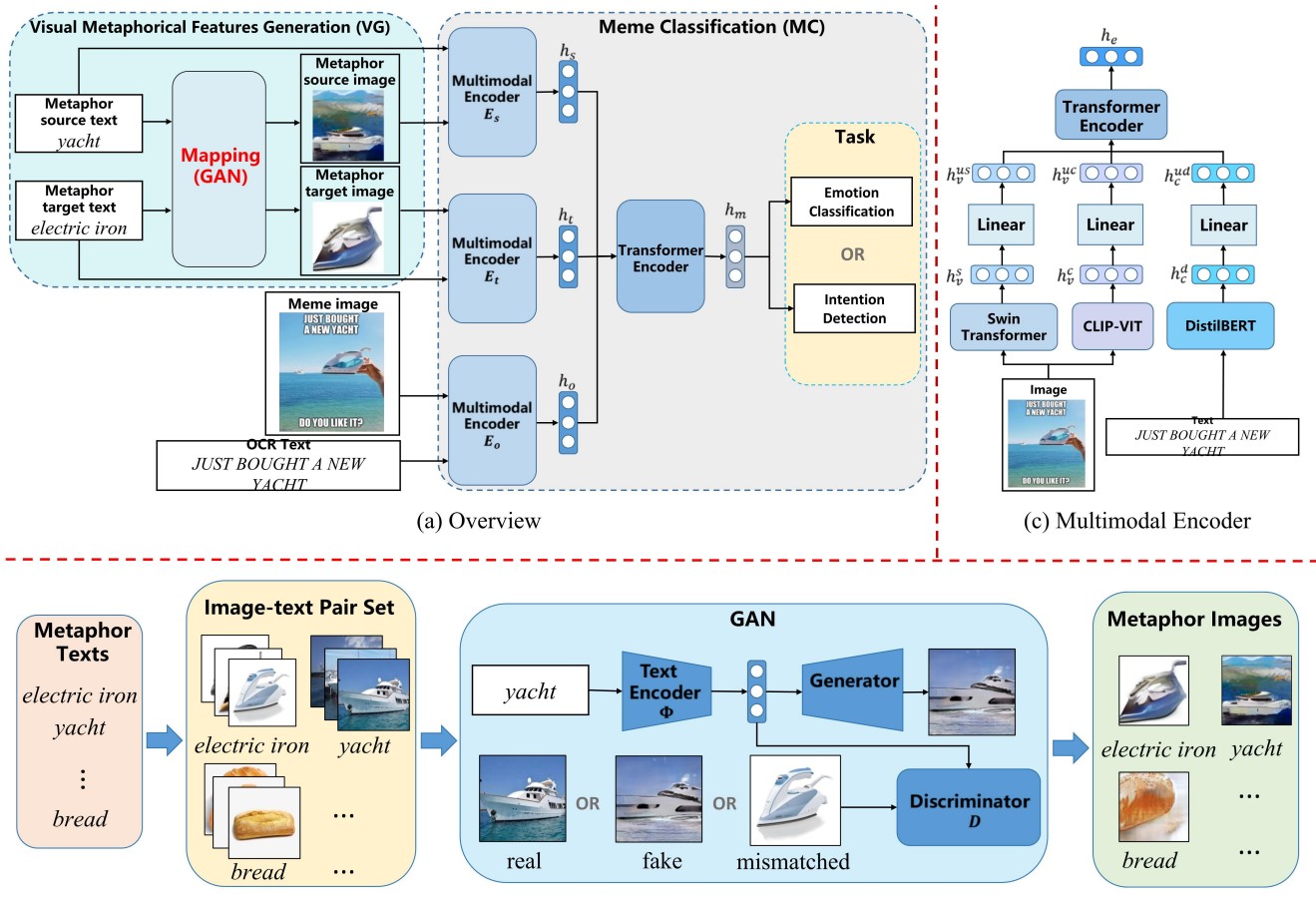

(a) Overview

(c) Multimodal Encoder

(b) Mapping(GAN)

**Figure 3: The structure of MMMC. In (a), the overview is presented, which contains two phases: Visual metaphorical feature Generation (VG) and Meme Classification (MC). In VG, metaphor texts are mapped into image space to generate metaphor images as visual metaphorical features. In MC, meme with its multimodal metaphorical features are fed into a classification model to predict its emotion or intention. Besides, (b) shows the detail about how we generate metaphor image via GAN, and (c) shows the structure of Multimodal Encoder(ME), which is used in MC.**

$D$, which is trained to classify if an image-text pair is real (i.e., from $S_k$) or fake (i.e., generated by $G_t$ or paired with wrong texts).

In this paper, we employ the prevalent BERT [6] as the text encoder which is pretrained on large amounts of textual data, and the architecture of RAT-GAN [31] as the design of $G_t$ and $D$, since RAT-GAN has demonstrated effectiveness particularly on text-to-image synthesis.

The generator $G_t$ is trained to generate "real" images as much as possible given an input text, while the discriminator $D$ is trained to distinguish real-world images and the generated images. Since the two models have the opposite goals, they are cross-optimized until convergence during training. The overall training objective of discriminator and generator can be formulated as:

$$\mathcal{L}_D = \mathbb{E}_{x \sim p_{data}}[max(0, 1 - D(x, \Phi(c)))]$$
$$+ \frac{1}{2}\mathbb{E}_{x \sim p_G}[max(0, 1 + D(x, \Phi(c)))] \quad (6)$$
$$+ \frac{1}{2}\mathbb{E}_{x \sim p_{data}}[max(0, 1 + D(x, \Phi(\hat{c}))]$$

$$\mathcal{L}_G = \mathbb{E}_{x \sim p_G}[min(D(x, \Phi(c)))] \quad (7)$$

Here, $c$ is the given text, $\hat{c}$ is a mismatched text, and $x$ is real or generated image. We use the trained model for image generation. For each $c \in U_k$, we keep 300 generated images, from which we manually select the most natural image, denoted as $v$, as the metaphor image of $c$.

We repeat the above procedure for all subsets $U_k$ ($k \in \{1, 2, \ldots, n_c/l\}$), and thereafter, obtain a multimodal metaphorical feature set

$$\Gamma = \{(v, c)|c \in U\} \quad (8)$$

where $v$ is the metaphor image corresponding to metaphor text $c$. We use this set to enrich the feature representation of each meme $m$, yielding a transformed representation denoted as $\hat{m}$.

## 4.2 Meme Classification(MC)

After obtaining the enriched representation $\hat{m}$, we combine and integrate the multimodal feature representation for classification tasks. As shown in Figure 3(a), we employ a dual-stage feature integration. In the first stage, the visual and text features are integrated for enhanced understanding of the metaphor source, metaphor target, and original meme, respectively. In the second stage, the embedded features of metaphor source, target, and original meme are fused for jointly use of classification. Next, we detail the two stages and the use of integrated features for classification.

*4.2.1 Multimodal Feature Integration.* For an image-text pair, we use a Multimodal Encoder to integrate the visual and textual features. In particular, we use three Multimodal Encoders(MEs) for the original meme, metaphor source, and metaphor target concept:

$$h_k = ME_k(k_v, k_c), \quad k \in \{o, s, t\} \tag{9}$$

As shown in Figure 3(c), given an image-text pair $(v, c)$, ME first extracts monomodal representations, and then fuses them into a multimodal representation. Similar to Tang et al. [28], we adopt Swin Transformer [18] and CLIP-ViT [22] to extract image features, and adopt DistillBERT [25] to extract text feature. Among them, Swin transformer is an image encoder that utilizes sliding windows and hierarchical structures, which is suitable for extracting visual representation of memes since many memes are composed of multiple sub-images; CLIP-ViT is a transferable vision model that uses text as supervised signal to train, which can extract visual features that are relevant on text; and DistilBERT is a text encoder that is a light version of the BERT model obtained through knowledge distillation. Here, the Swin Transformer pre-trained on ImageNet-21k [24] and the CLIP-ViT pre-trained on COCO [17] are adopted. During training, we fine-tune Swin Transformer, while freezing CLIP-ViT and DistilBert to reduce computational cost and improve the robustness of training. Given an input-text pair $(v, c)$, we denote the encoded feature of Swin Transformer as $h_v^s \in \mathbb{R}^{768 \times 1}$, CLIP-ViT as $h_v^c \in \mathbb{R}^{512 \times 1}$ and DistilBERT as $h_c^d \in \mathbb{R}^{768 \times 1}$. Then, we employ full-connected layers to map them onto the same dimension:

$$h_v^{us} = W_s h_v^s + b_s \tag{10}$$

$$h_v^{uc} = W_c h_v^c + b_c \tag{11}$$

$$h_c^{ud} = W_d h_c^d + b_d \tag{12}$$

Here, $W_s \in \mathbb{R}^{512 \times 768}$, $W_c \in \mathbb{R}^{512 \times 512}$, $W_d \in \mathbb{R}^{512 \times 768}$ are learnable parameters, $b_s, b_c$ and $b_d$ are learnable biases. Afterwards, uniformed feature vectors are fused by a transformer encoder as follows:

$$I = [h_v^{us}; h_v^{uc}; h_c^{ud}] \tag{13}$$

$$X_1 = MHA(I, I, I) \tag{14}$$

$$X_2 = Relu(W_2(W_1 X_1 + b_1) + b_2) \tag{15}$$

$$h_e = RowAvg(X_2) \tag{16}$$

where, $MHA$ is multi-head attention mechanism, $W_1$ and $W_2$ are two weight matrices, $b_1$ and $b_2$ are two learnable biases, and $h_e$ is the final output of ME.

| Task | Label | Amount | Proportion |
|------|-------|--------|------------|
| Emotion Classification | happiness | 1050 | 26.25% |
| | love | 693 | 17.32% |
| | anger | 567 | 14.17% |
| | sorrow | 583 | 14.57% |
| | fear | 123 | 3.07% |
| | hate | 701 | 17.52% |
| | surprise | 293 | 7.32% |
| Intention Detection | interactive | 220 | 5.50% |
| | expressive | 1296 | 32.40% |
| | purely entertaining | 1464 | 37.35% |
| | offensive | 1011 | 25.27% |
| | other | 9 | 0.22% |

**Table 1: Summary of class distribution of MET-meme.**

| Category | Metaphor Texts |
|----------|----------------|
| human organs | mouth, eyes, ear,figure,arm,lung,... |
| transportation | car, yacht, train, motorbike,bike,... |
| food | food, chips and gravy, hot dog,noodle,... |
| daily necessities | cup,knife,toilet paper,shampoo bottle... |
| clothing | clothes, a pair if socks, wedding dress,... |
| ... | ... |

**Table 2: Example metaphor texts extracted.**

*4.2.2 Feature Fusion.* After obtaining $h_o, h_s, h_t$, we fuse them through another transformer encoder. The same operations in Eq. (13)~(16) are performed for $h_o, h_s, h_t$, to obtain a final metaphorical meme representation $h_m$ that incorporates meme and its multimodal metaphorical information.

*4.2.3 Class Prediction.* Finally, $h_m$ is fed into a fully connected network with the softmax activation function for the final classification:

$$\hat{y} = \arg \max_y (softmax(W_m h_m)) \tag{17}$$

where $\hat{y}$ is the predicted label. Afterwards, $\hat{y}_i$ is used to calculate the categorical cross-entropy, which is used as the loss function:

$$\mathcal{L}_{cls} = -\frac{1}{N} \sum_{i=1}^{N} y_i \log \hat{y}_i \tag{18}$$

where $N$ is the number of samples, and $y_i$ is the true label of the $i^{th}$ sample.

## 5 EXPERIMENTS

### 5.1 Dataset

We conduct experiments on the English memes of MET-Meme [30] due to its richness in metaphorical memes. Specifically, MET-Meme contains 4,000 English memes, including 1,114 metaphorical memes and 2,886 non-metaphorical memes. For non-metaphorical memes, MET-Meme provides their images, OCR texts, emotion labels and intention labels. For metaphorical memes, MET-Meme provides additional metaphor source texts and metaphor target texts as shown in Figure 2. Table 1 summarizes the distribution of labels, which shows uneven distribution of classes.

## 5.2 Generation of Visual Features

*5.2.1 Training Details.* We extract metaphor concepts in MET-Meme, where we remove the duplicated concepts and those abstract concepts (e.g., "green life" and "Saturday"), of which images are difficult to obtain. As a result, we obtain a metaphor concept set $U$ of 293 unique metaphor concepts encompassing human organs, transportation, food, etc. Table 2 shows part of them. Then, we run the VG phase on $U$ with $l = 10$. Note that $U$ is divided into subsets so that one subset contains 13 metaphor texts. In addition, real images are collected from Google Images, Bing Images and Baidu Images. We train the GAN model for 3000 epochs with a batch size of 10 and two Adam optimizers [13] for generator and discriminator separately. The learning rates for generator and discriminator are set to $1e^{-4}$ and $4e^{-4}$, respectively.

*5.2.2 Generation Results.* After generation and selection, we obtain the multimodal metaphorical features set Γ, which contains 293 metaphor image-text pair. Figure 4 shows a part of them. We see that most metaphor images are natural and can reflect general appearance of metaphor concept. Hence, these images can be used as the visual features of metaphor concepts.

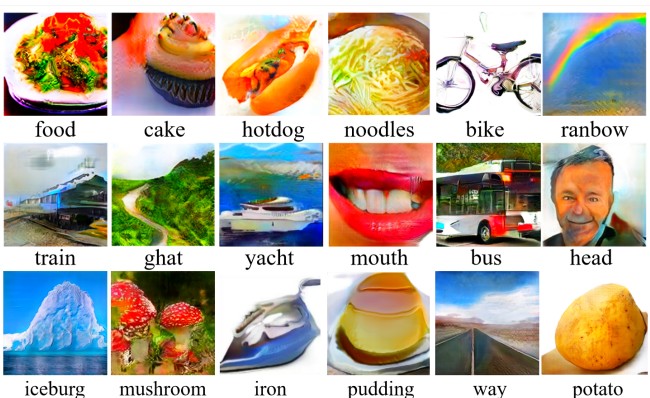

**Figure 4: Example of generated metaphor images given corresponding metaphor concepts.**

*5.2.3 Computational resources.* We utilize 3 2080Ti GPU to train GAN at a speed of 17 hours per repeat. And it takes 1.7 hours to generate a metaphorical image representing a specific concept. While this time investment is relatively significant, we consider it acceptable given that the process does not take up inference time and that a single metaphor image can be used for multiple modality understanding tasks.

## 5.3 Meme Classification

We split the 4000 English memes into training, validation and test sets with the ratio of 8:1:1. For missing metaphor texts, we set them as *"None"*. For their corresponding images, we set them as all-black images. An identical set of hyper-parameters is used for both emotion classification and intention detection tasks: a batch size of 50, dropout rate of 0.2 for each fully-connected layer, and an Adam optimizer with a learning rate of $1e^{-4}$. To avoid over-fitting,

We use early stopping if the loss on validation set does not get reduced within 500 epochs.

## 5.4 Baseline Models

We compare the proposed MMMC with existing meme classification models with demonstrated performance. These baseline methods are categorized into two groups: non-metaphorical methods and metaphorical methods. For non-metaphorical methods, we use methods of the teams that achieved high rankings at Memotion 2.0/3.0 competition[19, 23], including Yet[36], BLUE[2], Little Flower [21], NYCO-TWO [28], and the method of Ramamoorthy et al. [23]. For metaphorical methods, we use the method provided by Xu et al. [30], which uses only metaphor texts as additional metaphorical features. Additionally, we have adopted the state-of-the-art vision-language pre-trained model BLIP-2(7B)[16] as the benchmark for our study, which has demonstrated remarkable capabilities across a range of vision-language tasks. Notably, BLIP-2 undergoes refinement through fine-tuning. We implement all algorithms on a Nvidia RTX4090 GPU machine.

## 5.5 Results

We use weighted precision, weighted recall and weighted F1-value as evaluation metrics. The experimental results are reported in Table 3, which confirm that the proposed method manages to achieve consistent improvement in terms of all metrics compared with other methods on both emotion classification and intention detection. In addition, we conduct pairwise *t*-test on weighted F1-value comparing MMMC with NYCO-TWO and the method of Xu et al. All of the produced *p*-values are less than 0.05. According these results, we can draw the following two preliminary conclusions.

First, extra metaphorical information improve meme understanding. The experimental results on both classification tasks confirm that metaphorical methods are more competitive than non-metaphorical methods. On both tasks, our MMMC and the method of Xu et al. obtain the highest and second highest weighted F1-value. In addition, NYCO-TWO performs the best among non-metaphorical methods and significantly outperforms other non-metaphorical methods.

Second, multimodal metaphorical features improve meme understanding. We observe that MMMC outperforms the method of Xu et al., which only uses linguistic representation of metaphor. Compared with the method of Xu et al., MMMC achieves a 1.14% improvement in weighted F1-value on emotion classification and a 6.54% improvement on intention detection.

To further analyze the classification performance of MMMC, we present the confusion matrix in Figure 5. On emotion classification, we observe that model has a high recall in *happiness*. In 115 happiness memes, 77 are predicted correctly. However, we also notice that all categories of memes are more likely be predicted as *happiness*. For example, in 53 *sorrow* memes, only 11 are correctly predicted, while 23 are mispredicted as *happiness*. In addition, model performs poorly in recognizing *fear* and *surprise*. To be specific, none of the 11 *fear* memes and 29 *surprise* memes are correctly predicted, which may be caused by the imbalanced label distribution in dataset. Similarly, on intention detection, model performs poorly in recognizing *interactive* and *other* due to the small number of

| Methods | Metaphor text | Metaphor image | Emotion Classification | | | Intention Detection | | |
|---|---|---|---|---|---|---|---|---|
| | | | P | R | F1 | P | R | F1 |
| Little Flower | - | - | 26.75 | 26.31 | 26.01 | 36.43 | 38.25 | 37.28 |
| YET | - | - | 26.38 | 29.50 | 27.81 | 41.79 | 43.75 | 42.48 |
| Ramamoorthy et al. | - | - | 30.30 | 30.75 | 30.42 | 42.29 | 43.50 | 41.50 |
| BLUE | - | - | 28.73 | 33.00 | 30.29 | 43.26 | 43.00 | 42.34 |
| NYCO-TWO | - | - | 31.24 | 35.50 | 31.59 | 44.73 | 46.75 | 45.57 |
| BLIP-2 | + | - | 32.94 | 30.75 | 30.61 | 43.01 | 44.75 | 43.66 |
| Xu et al. | + | - | 31.51 | 35.75 | 32.76 | 47.50 | 45.85 | 46.05 |
| MMMC(ours) | + | + | **33.22** | **37.25** | **33.90** | **52.95** | **54.00** | **52.59** |
| w/o metaphor image | + | - | 31.33 | 34.75 | 32.39 | 49.75 | 51.50 | 50.24 |
| w/o metaphor text | - | + | 33.10 | 36.75 | 33.04 | 50.54 | 52.00 | 50.64 |
| w/o metaphor text and image | - | - | 31.06 | 33.75 | 31.84 | 48.21 | 49.50 | 48.00 |

**Table 3: Results on MET-Meme. "+" represents that the corresponding feature is used by the method.**

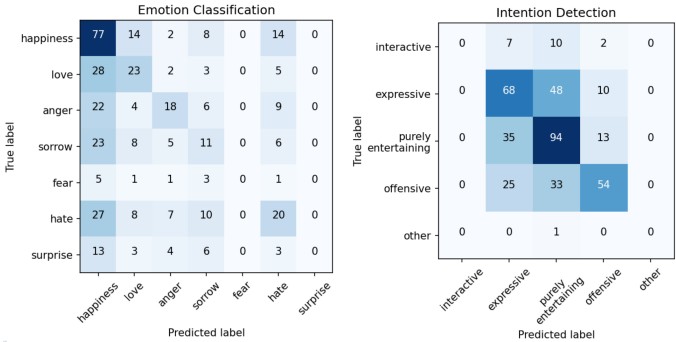

**Figure 5: Confusion matrix of results on MMMC.**

samples. On the other hand, model performs well in distinguishing *expressive*, *purely entertaining*, and *offensive*, where more than half of them are correctly classified.

## 5.6 Qualitative Analysis

*5.6.1 Feature Ablation.* We further evaluate MMMC considering following scenarios: (1) without metaphor text; (2) without metaphor image; and (3) without both metaphor text and image. For w/o metaphor text, we drop $h_c^{ud}$ in Eq. (13). For w/o metaphor metaphor image, we replace $ME_s$ and $ME_t$ with DistilBert. For w/o metaphor text and image, we use $ME_m$ to implement classification directly. Table 3 reports the results. We observe that the worst performance is in the case of without any metaphorical features. Results can be improved by adding metaphorical features of any mode. Hence, we determine that extra metaphorical information can improve meme understanding. In addition, we observe that the case of adding multimodal metaphorical features outperforms the case of adding monomodal metaphorical feature. In particular, compared to w/o metaphor text and w/o metaphor image, our multimodal MMMC achieves a 0.86% improvement and a 1.51% improvement in weighted F1-value on emotion classification. Similarly, the performance improvement of intent detection is 1.95% and 2.35%. This confirms that multimodal metaphorical features can improve meme

understanding. We also observe that visual metaphorical feature is more important than textual metaphorical feature. Specifically, w/o metaphor text surpasses the performance of w/o metaphor image by 0.65% and 0.40% F1-score on emotion classification and intention detection. Moreover, our *ME* is effective since the case of w/o metaphor text and image outperforms all non-metaphorical methods, and even outperforms the metaphorical method of Xu et al. on intention detection.

*5.6.2 Fusion Strategy.* To demonstrate the effectiveness of our fusion strategy involving TransformerEncoder(TE) and dual-stage feature integration, we conduct evaluations for MMMC under following scenarios: (1) single-stage fusion with concatenation, (2) single-stage fusion with TE, and (3) dual-stage fusion with concatenation. In the single-stage scenarios, we incorporate all features together. In the concatenation scenarios, we replace TE with the concatenation operation. Table 4 shows the results of our experiments. We observe that the use of TE instead of concatenation operation yields high enhancement. This indicates that TE is able to capture the inter-feature relationships more effectively. In addition, dual-stage feature integration outperforms single-stage feature integration when using the same fusion method. This suggests that dual-stage feature integration can more effectively utilize the information from different stages, resulting in improved performance in meme understanding.

| Fusion Method | | Emotion Classification | | | Intention Detection | | |
|---|---|---|---|---|---|---|---|
| | | P | R | F1 | P | R | F1 |
| Single | cat | 31.88 | 34.75 | 32.47 | 48.78 | 51.75 | 49.90 |
| | TE | 32.63 | 37.25 | 33.58 | 50.78 | 53.75 | 51.28 |
| Dual | cat | 32.32 | 35.75 | 32.81 | 51.31 | 53.00 | 50.88 |

**Table 4: Results obtained using different fusion strategies.**

*5.6.3 Parameter sharing.* To explore the feasibility of parameter sharing among different $ME_k$ modules, we conduct additional evaluations. Our study encompass the following scenarios: (1) $ME_s$ shares parameters with $ME_t$, while $ME_o$ does not share parameters; (2) Parameters are shared among $ME_s$, $ME_t$, and $ME_o$; (3)

None of the modules share parameters. Table 5 displays the results obtained from our experiments. We observe that the most favorable outcomes are attained when parameter sharing is not employed among the three $ME_k$ modules. Conversely, the poorest results are obtained when all three $ME_k$ modules share parameters. This finding can be attributed to the inherent divergence in semantic interpretations between the source and target concepts of metaphorical expressions. Consequently, our methodology deliberately avoids parameter sharing to account for these semantic discrepancies more effectively.

| Parameter Sharing | | | Emotion Classification | | | Intention Detection | | |
|---|---|---|---|---|---|---|---|---|
| $ME_s$ | $ME_t$ | $ME_o$ | P | R | F1 | P | R | F1 |
| ✓ | ✓ | ✓ | 30.89 | 35.25 | 32.56 | 48.61 | 50.75 | 49.17 |
| ✓ | ✓ | ✗ | 33.08 | 36.00 | 33.03 | 52.44 | 53.75 | 51.72 |
| ✗ | ✗ | ✗ | **33.22** | **37.25** | **33.90** | **52.95** | **54.00** | **52.59** |

**Table 5: Results obtained using different parameter sharing settings. "✓" means that the parameters are shared.**

*5.6.4 Case Study.* In Figure 6, we present a case study. The meme contains metaphor message "*train is sweat*", and conveys happy emotion and offensive intention. For this meme, NYCO-TWO fails to predict both its emotion and intention. Emotion is mispredicted as *love* and intention is mispredicted as *expressive*. With additional linguistic features, the method of Xu et al. correctly predicts its emotion but mispredicts its intention to *purely entertaining*. Different from them, MMMC correctly predicts both its emotion and intention using multimodal metaphorical features. This shows that with the addition and enrichment of metaphorical features, the understanding of this meme is gradually enhanced.

| Meme Image | Metaphor | Ground Truth |
|---|---|---|
| | "*train is sweat*" source: *sweat* target: *train* | *happiness* and *offensive* |
| **NYCO-TWO** | **Xu et al.** | **our MMMC** |
| *love* and *expressive* | *happiness* and *purely entertaining* | *happiness* and *offensive* |

**Figure 6: Classification results comparison of our MMMC with NYCO-TWO and the method of Xu et al.**

## 6 CONCLUSION AND LIMITATION

In this paper, we proposed a novel meme classification method MMMC that generates multimodal metaphorical features to improve meme understanding. We leverage a text-conditioned GAN, and generate visual metaphorical features by mapping textul metaphor data into visual feature space. Furtherly, we design a classification model that incorporates meme and its multimodal metaphorical features. We perform our method MMMC on a MET-Meme dataset, which contains adequate metaphorical memes. Experimental results show that MMMC significantly outperforms other existing baselines.

Our work has some limitations. MMMC needs human-annotated metaphor texts to generate visual metaphorical features. However, we can not obtain human-annotated metaphor texts when given an unknown meme. Therefore, we plan to leverage a method that can automatically detect metaphor texts in future work. Furthermore, the analysis of memes is inextricably linked to their broader context, including cultural backgrounds and the associated posts. Thus, our future work includes refining how to embed this context into meme understanding.

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
