# OpenReview forum: "Generating Multimodal Metaphorical Features for Meme Understanding"
_acmmm.org/ACMMM/2024/Conference — MM2024 Oral_

### Official Review · Reviewer_48Na · 2024-05-07

**Rating:** 5
**Confidence:** 3

**Summary:**

This paper introduces a novel feature (visual metaphors) into the meme classification task and demonstrates improvement in utilising multimodal features as opposed to text-based features. It utilises a GAN-based approach for object synthesis to perform representation mapping to identify metaphor imagery. This is an interesting approach to enriching the visual-language domain with knowledge for implicit reasoning which are prevalent in the understanding of meme content. The authors derived a baseline modelling approach and performed ablation studies to provide empirical evidence to support the importance of the novel features, architecture design and strategies.

**Strengths:**

Strengths of the paper:
- Identification of novel features to improve multimodal understanding of meme content. Looking beyond the literal visual imagery to infer second-order visual-text mismatch to derive inherent emotions and intentions which are non-obvious.
- Has potential to incorporate social science theories on literary analysis for memetic content similar to analogies and sacarsm tasks for visual metaphors.
- Harnesses the strength of adversarial training to capture rich visual semantics for mapping framework, thereby allowing a stronger interaction just by using simple fusion architectures i.e. concatenation, single-stage, dual-stage etc. to achieve improvements.
- Paves the way forward for further research into emotive understanding of complex visual content to identify possible cultural references, and subtle narrative nuances for a difficult problem in automated social media content analysis and understanding.

**Limitations:**

There are a couple of points that I would need to address in the paper to clarify and hopefully, improve the paper's standing:

- What is the computational resources for Visual Metaphorical Feature generation? The paper did not highlight in terms of overheads for the feature generation from the metaphor concepts to obtain the representation for the multimodal encoder. While the evaluation shows a +2-7% improvement on downstream tasks for emotion classification and intent detection, it is difficult to appreciate the extent of improvement if the end-to-end framework requires a significant processing time for inference.

- The metaphorical categories are manually collected to reflect real-world usage of the content in memes, however there is no study on the effectiveness of the types of metaphorical content used. I believe the representations changes from culture to culture, and location as well as context plays an important role in this as well. Has this been effectively studied? In addition, how do the authors envisage this framework to scale and generalise across different meme contexts. How do we automatically enrich the metaphor categories and texts to reflect and capture social and cultural norms.

- In terms of architecture, the metaphor source and target images are considered "local-context" representations which are fed into the MEs for feature extraction, while the meme image is considered a "global-context" which is encoded by MEo. While the Swin transformer establishes a hierarchical structure for feature extraction, the alignment between local and global seems to be weak here. Is there any consideration to incorporate object-level semantics as most of the metaphorical features are related to objects rather than whole images in general?

- Under section 5.6.4 "Generalizability" - I thought this section was to address the generalizability of the model to various data domains rather than model structures (as this is the general consensus for model generalizability)? I think it would be more attractive to analyse domain distribution changes and how such metaphorical features introduces any form of invariance to such shifts.

- Additional points on paper formatting. There are a couple of spelling mistakes that needs to be fixed i.e. "Swin" not "Swim" transformers, "multimode" etc.

**Suitability:**

2

---

### Official Review · Reviewer_rzNN · 2024-05-08

**Rating:** 5
**Confidence:** 3

**Summary:**

Memes often incorporate metaphorical elements, necessitating a model's comprehension of both textual and visual metaphors. While previous studies typically focus solely on textual metaphors, this paper introduces a model that integrates metaphorical images. Utilizing the MET-Meme dataset, this dataset has human annotated metaphorical text for each meme. The authors initially train a Generative Adversarial Network (GAN) to generate metaphorical images based on input metaphorical text. Subsequently, the model performs meme classification by combining original text, original images, metaphorical text, and metaphorical images. Experimental results demonstrate the superior performance achieved by including metaphorical images, highlighting the significance of both textual and visual metaphors in meme comprehension.

**Strengths:**

1. Performance: This paper introduces a model that significantly surpasses state-of-the-art models in meme classification.
2. Novelty: This paper pioneers the utilization of metaphorical images in meme classification, highlighting the crucial role of metaphorical information, particularly metaphorical images, in understanding memes.
3. Clarity: The clarity and accessibility of this paper's writing are highly commendable.

**Limitations:**

1. Model Construction: Why utilize a GAN model to generate metaphorical images for each textual metaphor instead of directly using online memes(select the one with highest similarity to the textual metaphors)? Could this approach impact model performance?
2. Applications: Due to its reliance on annotated metaphor text, this paper is limited to the MET-Meme dataset and cannot demonstrate its superior performance across other datasets.
3. Insufficient Evaluation: While the dataset comprises metaphor source text and metaphor target text, indicating a distinction between the two, the paper does not delve into their respective influences on the model.

**Suitability:**

3

---

### Official Review · Reviewer_CdPJ · 2024-05-13

**Rating:** 3
**Confidence:** 2

**Summary:**

This paper proposes a method introducing metaphorical labels into the model for the meme classification task, named MMMC. It uses real images collected from the Internet to train a GAN to generate visual features of certain metaphor concepts, and then use multi-head attention to fuse these features for classification. Experiments on the MET-Meme dataset show the improvement of this method.

**Strengths:**

- The code is published, which means it is easy to reproduce.
- Experiments show that this method outperforms the baselines in the current evaluation framework.
- The absolution studies are sufficient.

**Limitations:**

1.  I wonder whether BLIP-2 in Table 3 is fine-tuned on MET-Meme. If not, it seems to be unfair to BLIP-2.
2.  It would be better to compare MMMC with some multimodal large language models (larger than BLIP-2) , e.g., Flamingo[1] and GPT-4[3], given that some studies [2,3,4] find that multimodal large language models can understand metaphors even in zero-shot or few-shot learning.
3.  Some writing suggestions:

- Reference [34] appears to have a formatting error.
- In line 339, "inputc" -> "input c".

> [1] Jean-Baptiste Alayrac, Jeff Donahue, Pauline Luc, Antoine Miech, Iain Barr, Yana Hasson, Karel Lenc, Arthur Mensch, Katherine Millican, Malcolm Reynolds, Roman Ring, Eliza Rutherford, Serkan Cabi, Tengda Han, Zhitao Gong, Sina Samangooei, Marianne Monteiro, Jacob L Menick, Sebastian Borgeaud, Andy Brock, Aida Nematzadeh, Sahand Sharifzadeh, Mikołaj Bińkowski, Ricardo Barreira, Oriol Vinyals, Andrew Zisserman, and Karén Simonyan. 2022. Flamingo: a Visual Language Model for Few-Shot Learning. In Advances in Neural Information Processing Systems, 23716–23736.
>
> [2] EunJeong Hwang and Vered Shwartz. 2023. MemeCap: A Dataset for Captioning and Interpreting Memes. In Proceedings of the 2023 Conference on Empirical Methods in Natural Language Processing, Singapore, 1433–1445.
>
> [3] OpenAI. 2023. GPT-4 Technical Report. CoRR abs/2303.08774, (2023).
>
> [4] Linhao Zhang, Li Jin, Guangluan Xu, Xiaoyu Li, Cai Xu, Kaiwen Wei, Nayu Liu, and Haonan Liu. 2024. CAMEL: Capturing Metaphorical Alignment with Context Disentangling for Multimodal Emotion Recognition. Proceedings of the AAAI Conference on Artificial Intelligence 38, 8 (2024), 9341–9349.

**Suitability:**

3

---

### Official Review · Reviewer_SEUH · 2024-05-25

**Rating:** 4
**Confidence:** 2

**Summary:**

This paper introduces MMMC, a framework to generate metaphorical features in memes for better meme understanding. MMMC uses GANs to create visual metaphors, which are fed into classifiers for downstream tasks.

**Strengths:**

1. The finding that visual metaphors are better than text metaphors is interesting.
2. MMMC uses multimode encoders with multiple representation learners for each modality.
3. Visual metaphors reamin unexplored in this domain but are crucial for creative meme understanding and content moderation.
4. A GAN is intuitively used in MMMC for visual metaphor generation, boosting meme understanding for downstream classifiers.
5. The comparison with previous methods is sound and highlights the contribution of each aspect of MMMC towards its performance.

**Limitations:**

1. With generative models for meme understanding, this framework may be very computationally expensive for real-world usage. This paper could include more details about this.
2. While the visual metaphors generated by the GAN are decent, wouldn't diffusion models create much better metaphors?
3. The disconnect between emotion classification and intent detection, i.e. how most models are closer to MMMC for emotion classification than intent detection, could be explained better.

**Suitability:**

3

---

### Meta-Review · Area_Chair_cfHN · 2024-06-29

**Recommendation:** Accept (Oral)
**Confidence:** 5

**Metareview:**

The manuscript presents a generative framework for meme understanding using GAN models to create visual metaphors. The paper's strengths include novel features that improve the multimodal understanding of meme content, the potential to incorporate social science theories, and the ability to harness adversarial training to capture rich visual semantics. The model significantly surpasses SOTA's performance in meme classification and pioneers the use of metaphorical images, with the paper's clarity being highly commendable. However, several limitations need addressing. The framework's high computational expense requires detailed analysis, and diffusion models might yield better visual metaphors. The disconnect between emotion classification and intent detection needs clarification, and fairness in the BLIP-2 comparison should be examined, including evaluation with larger multimodal models like Flamingo and GPT-4. The reliance on annotated metaphor text restricts applicability to the MET-Meme dataset, and the significant time required for generating metaphorical images raises sustainability concerns. Authors are asked to address these concerns in the camera-ready version.